# FCN-LLM: Empower LLM for Brain Functional Connectivity Network Understanding via Graph-level Multi-task Instruction Tuning

## Abstract

Large Language Models (LLMs) have achieved remarkable success in language understanding and reasoning, and their multimodal extensions enable comprehension of images, video, and audio. Inspired by this, foundation models for brain functional connectivity networks (FCNs) derived from resting-state fMRI have shown promise in clinical tasks. However, existing methods do not align FCNs with the text modality, limiting the ability of LLMs to directly understand FCNs. To address this, we propose FCN-LLM, a framework that enables LLMs to understand FCNs through graph-level, multi-task instruction tuning. Our approach employs a multi-scale FCN encoder capturing brain-region, functional subnetwork, and whole-brain features, projecting them into the LLM's semantic space. We design multi-paradigm instruction tasks covering 19 subject-specific attributes across demographics, phenotypes, and psychiatric conditions. A multi-stage learning strategy first aligns FCN embeddings with the LLM and then jointly fine-tunes the entire model to capture high-level semantic information. Experiments on a large-scale, multi-site FCN database show that FCN-LLM achieves strong zero-shot generalization on unseen datasets, outperforming conventional supervised and foundation models. This work introduces a new paradigm for integrating brain functional networks with LLMs, offering a flexible and interpretable framework for neuroscience.

## 1 Introduction

In recent years, the rapid advancement of Large Language Models (LLMs) has led to remarkable breakthroughs in their ability to comprehend language (Devlin et al., 2019; Chen et al., 2024a), generate coherent dialogues (Floridi & Chiriatti, 2020; Achiam et al., 2023), and perform complex text-based reasoning (Guo et al., 2025). However, their reliance on text-only inputs poses fundamental limitations in perceiving and reasoning over the rich variety of information humans naturally process. To address these limitations, the research community has focused on aligning other modalities, such as image, video, and speech, into the semantic space of LLMs, where these proposed models are recognized as Large Multimodal Models (LMMs) (Liu et al., 2023; Li et al., 2023; Lin et al., 2023; Boson AI, 2025). Through alignment between other modalities and the text, LMMs can extract, understand, and reason about information from non-textual data, as well as generating responses in a textual format (Han et al., 2023; Wang et al., 2024c). Such multimodal extensions of LLMs have achieved strong generalizability and state-of-the-art performance across a broad range of downstream tasks, including visual question answering (Antol et al., 2015; Marino et al., 2019), image captioning (Lin et al., 2014), video reasoning (Fu et al., 2025), audio transcription (Panayotov et al., 2015), and speech-grounded dialogue (Wang et al., 2024a). Altogether, by holding text as mainly input and output, LMMs offer a powerful new paradigm for artificial intelligence with exceptional flexibility, intuitive interpretability, and natural interactivity with humans.

Meanwhile, the successes of LLMs, underpinned by large-scale pre-training and the scalable Transformer-based model architecture (Vaswani et al., 2017), have inspired the neuroscience community to explore analogous foundational models for various modalities of neuroimaging data (Yang et al., 2024b; Hu et al., 2025; Wei et al., 2025). The goal is to learn robust and generalizable representations that can be transferred to diverse downstream tasks, which is crucial for clinical translation. In

the context of brain functional connectivity networks (FCNs) derived from resting-state functional Magnetic Resonance Imaging (rs-fMRI), such emerging foundation models have been proposed for understanding brain function (Power et al., 2011) and supporting clinical applications such as disease diagnosis (Yang et al., 2021; Wang et al., 2024d) and outcome prediction (Finn et al., 2015; He et al., 2024), recent efforts have moved beyond graph-based supervised deep learning approaches (Li et al., 2021; Kawahara et al., 2017; Kan et al., 2022b). Remarkably, several studies (Hu et al., 2024; Yang et al., 2024b; Wei et al., 2025) have begun to aggregate large-scale, multi-site FCN datasets and to apply different pretraining strategies to build foundation models for improved performance. These foundation models have demonstrated promising generalization by extracting transferable features from FCNs and achieving strong performance across multiple downstream clinical tasks.

Despite these advances, a critical gap remains: existing methods fail to align functional connectivity networks (FCNs) with text. This blocks multimodal large language models (MLLMs)—which have firmly established semantic space as the universal foundation for multimodal understanding—from directly interpreting the rich brain-related information in FCNs. A key barrier is the absence of a systematic language-based framework for FCNs: this stands in sharp contrast to the vision domain, where images naturally pair with text for contrastive pretraining (Radford et al., 2021) or caption generation (Li et al., 2022; Yu et al., 2022), and prevents the transfer of proven multimodal alignment strategies to FCNs—even though FCNs inherently contain sufficient text-relevant representations to support language-based multiscale decoding of universal FCN embeddings. Compounding this limitation, current FCN foundation models still rely on task-specific fine-tuning (using downstream data) or separate classifiers (e.g., linear support vector machine (SVM)) for each task. As a result, they fail to achieve genuine zero-shot generalization, as they cannot leverage cross-modal knowledge transfer through MLLMs' semantic space and the latent text-compatible representations within FCNs.

To explore to what extent FCNs can be aligned with the text modality and to address the intriguing question of *whether LLMs can understand FCNs*, we propose FCN-LLM, a framework that empowers LLMs to understand FCNs through graph-level, multi-task instruction tuning. Specifically, we first design a **multi-scale FCN encoder** that extracts features from three complementary levels of FCNs—brain region-of-interest (ROI) level, functional subnetwork level (Yeo et al., 2011), and the whole brain level—and projects them into the semantic space of the LLM via a Multilayer Perceptron (MLP). Second, given the inherent difficulty of describing FCNs in natural language, from the perspective of graph-level pretraining (Hu et al., 2019), we develop **multi-paradigm task design for instruction tuning** which requires predicting domain-specific attributes of the subjects corresponding to FCNs. In total, we curate 19 attributes spanning demographics, phenotypes, and psychiatric conditions, and synthesize a large amount of instruction-answer pairs across three paradigms—predictive, judgment, and comparative—to guide model training. Third, we introduce a **multi-stage learning of FCN knowledge**: in the first stage, we only train the multi-scale FCN encoder to align FCN embeddings with the LLM's semantic space; and in the second stage, we jointly fine-tune the entire model to capture high-level and abstract semantic information of FCNs.

We collect a large-scale, multi-site FCN database for training FCN-LLM and conduct comprehensive comparisons against both conventional supervised FCN models and recently developed FCN foundation models. The experimental results show that while FCN-LLM performs slightly worse than the supervised FCN models on in-domain test sets, it achieves state-of-the-art generalization performance when tested on unseen datasets in a zero-shot manner. This strong generalizability is attributed to two key factors: the graph-level, multi-task instruction tuning that endows the FCN features with excellent generalization properties, and the inherent flexibility provided by LLM powerful language comprehension and generation abilities. We also provides interpretability by using prompt-conditioned attention maps to highlight subnetwork-level FCN connections as potential brain biomarkers, as shown in Appendix A.7.

## 2 RELATED WORK

### 2.1 FOUNDATION MODELS FOR BRAIN FUNCTIONAL CONNECTIVITY NETWORK ANALYSIS

Foundation models, pretrained on large-scale datasets through self-supervised learning, exhibit remarkable transferability across a wide range of downstream tasks. For foundation models on FCNs, several pre-training strategies have been explored. For example, BrainNPT introduced a replaced-ROI prediction strategy and employed the [CLS] token embedding for downstream applications (Hu et al.,

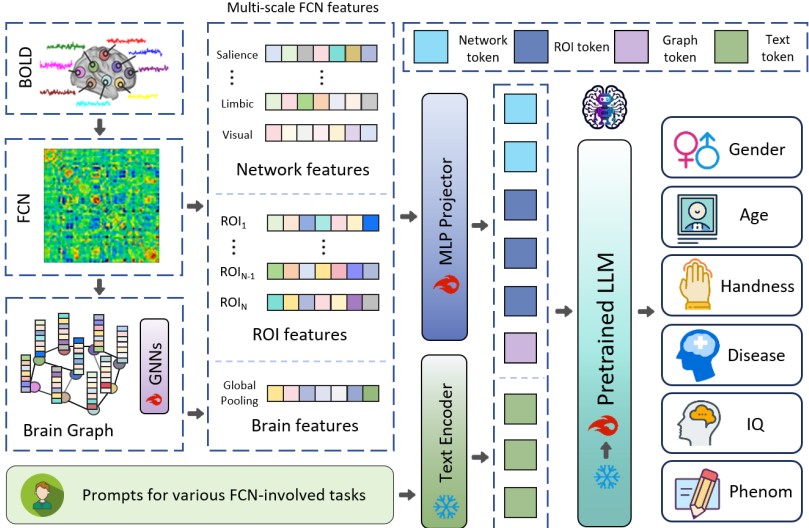

Figure 1: The model architecture of our proposed FCN-LLM that supports queries related to various FCN-involved tasks. It employs multi-scale encoder to extract FCN features, maps these features into the text embedding space.

2024). BrainMass adopted mask-ROI modeling and latent representation alignment to learn robust representations, followed by linear probing for evaluation (Yang et al., 2024b). CINP performed contrastive pre-training between FCNs and structural MRI to construct a joint semantic space (Hu et al., 2025). BrainGFM leveraged graph contrastive learning and graph masked autoencoders to pre-train on a diverse set of brain atlases with varying parcellations, thereby improving the model's generalization and adaptability across heterogeneous fMRI-derived brain representations (Wei et al., 2025). Although these FCN foundation models have demonstrated strong transferability across multiple downstream tasks, they still suffer from two key limitations, i.e., the lack of alignment with the LLM's semantic space and the reliance on task-specific fine-tuning for effective deployment.

## 2.2 ALIGNMENT FOR NEW MODALITIES TO LARGE LANGUAGE MODELS

Recent research has made substantial progress in aligning new modalities (e.g., images and videos) with LLMs. One prominent line of work introduces projection modules that map visual features into the input space of LLMs. For example, BLIP-2 and MiniGPT-4 employed a Q-Former that interacts with image features via cross-attention before projecting them into the LLM's input space (Li et al., 2023; Zhu et al., 2023). LLaVA exploited a lightweight yet effective MLP to map features from a CLIP visual encoder into visual tokens (Liu et al., 2023). Another line of research focuses on multi-modal instruction tuning and unified embeddings. ImageBind-LLM (Han et al., 2023), for instance, was trained solely on image-text alignment using a "bind network" and gating layers, yet generalized seamlessly to video, audio, and 3D modalities by leveraging embeddings from ImageBind. Video-LLaVA (Lin et al., 2023) further advanced this direction by learning a unified representation for images and videos, enabling LLMs to process both modalities within a shared semantic space. Inspired by the efficiency of projection-based alignment modules, in this paper we adopt an MLP to map FCN features into the input space of LLMs for enhancing the multi-modality alignment of our FCN-LLM.

## 2.3 MULTI-TASK PROMPT TUNING

Multi-task prompt tuning has emerged as an effective scheme for adapting LLMs to many downstream tasks. It enables knowledge sharing by transferring useful patterns across related tasks, which improves efficiency and reduces redundancy (Asai et al., 2022; Wang et al., 2023). It also enhances generalization, as exposure to diverse tasks encourages more robust and universal representations that adapt better to previously unseen tasks (Shen et al., 2024). Moreover, multi-task prompt tuning provides flexibility, since new tasks can be supported by simply composing or extending lightweight

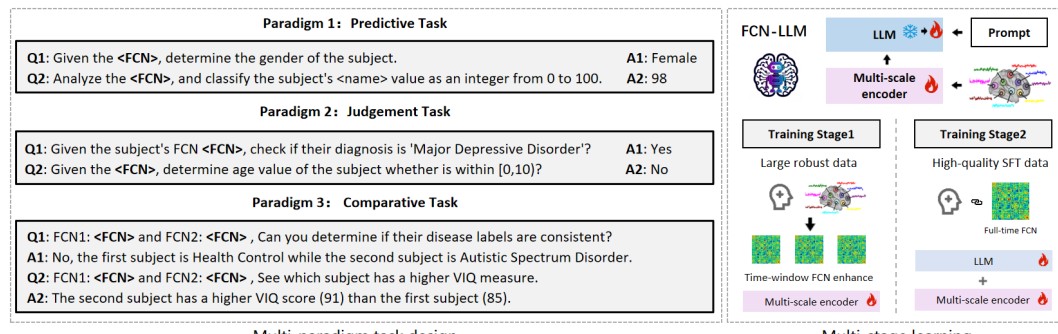

Figure 2: Illustration of the proposed multi-paradigm task design for instruction tuning (left), and the multi-stage learning of FCN knowledge learning for our proposed FCN-LLM (right).

prompts without retraining the full model (Wang et al., 2023). Finally, multi-task prompt tuning is particularly effective under both few-shot and zero-shot settings, where shared prompts stabilize learning and improve performance compared to single-task prompt tuning (Sanh et al., 2021; Asai et al., 2022; Shen et al., 2024). Considering these advantages, we adopt multi-task prompt tuning in this work to construct diverse instruction-tuning data and effectively transfer FCN knowledge to the LLM, thereby improving both generalization and robustness.

## 3 METHODOLOGY

This section is organized as follows. **(1) We introduce the architecture of FCN-LLM** (as illustrated in Figure 1), which is composed of two core components: a) the multi-scale FCN encoder and b) the pre-trained LLM. Specifically, FCNs are first converted into concrete feature embeddings via the multi-scale FCN encoder (see Sec. 3.1 for details); an MLP then projects and aligns these embeddings into the semantic space of the LLM and the LLM finally extracts information from FCNs to generate texts to accompish downstream tasks. The projection-based alignment used in FCN-LLM is not only intuitive but also efficient according to the successful experience of several LMMs (Liu et al., 2023; Li et al., 2023; Zhu et al., 2023). **(2) We propose a multi-paradigm task design for instruction tuning** in Sec. 3.2, which comprises three paradigms: predictive, judgment, and comparative, as displayed in the left panel of Figure 2. **(3) We construct a two-stage framework for FCN knowledge learning** in Sec. 3.3, as shown in the right panel of Figure 2, which consists of a pretraining stage for robust FCN knowledge alignment and a fine-tuning stage for high-level FCN knowledge refinement.

### 3.1 MULTI-SCALE FCN ENCODER

**Brain Region Level.** An FCN derived from rs-fMRI is generally characterized by a matrix (hereinafter called FCN matrix), whose elements denote functional connectivities (FCs) that are quantified as the dependence between blood oxygen level dependent (BOLD) time series of paired ROIs. We regard the rows of the FCN matrix as ROI-level feature vectors, each of which reflects FC patterns between one ROI and the others on a local scale. Let $D$ be the number of ROIs, and an ROI-level feature vector can be denoted by $\mathbf{f}_{\text{roi}} \in \mathbb{R}^D$.

**Functional Subnetwork Level.** Researchers have systematically partitioned the human cerebral cortex into seven discrete functional subnetworks through cluster analysis of a large amount of rs-fMRI data (Yeo et al., 2011). The subnetwork partition of the brain integrates the ROIs that are closely related in some function. Numerous studies have demonstrated the validity and necessity of this functional parcellation by correlating these functional subnetworks with mental disorders, phenotypes, and behavior (Ereira et al., 2024; Sheffield et al., 2015; Menon, 2023). To fully exploit this prior knowledge, we average the ROI-level features of those ROIs within each functional subnetwork to obtain subnetwork-level feature vectors.

Let $\mathcal{S}_k$ denote the index set of the ROIs belonging to the $k$-th subnetwork ($k = 1, 2, ..., N$), where $N$ is the number of subnetworks and $\mathbf{f}_{\text{roi},i}$ represent the ROI-level feature vector of the $i$-th ROI. The

feature vector of the $k$-th subnetwork is calculated as

$$\mathbf{f}_{\text{sub},k} = \frac{1}{|\mathcal{S}_k|} \sum_{i \in \mathcal{S}_k} \mathbf{f}_{\text{roi},i}. \tag{1}$$

where $|\mathcal{S}_k|$ stands for the cardinality of $\mathcal{S}_k$, i.e., the number of ROIs in the $k$-th subnetwork.

**Whole Brain Level.**  Although FCN information can be decoded at the fine-grained ROI and subnetwork levels, it is equally crucial to learn representations that capture whole-brain organization by leveraging the prior connectivity among brain regions, since large-scale network interactions play a central role in shaping brain function (Bullmore & Sporns, 2009). To this end, we first apply absolute value transformation and thresholding to the FCN to obtain an adjacency matrix $\mathbf{A} \in \mathbb{R}^{D \times D}$ that models the brain as graph structural data, where the nodes are ROIs, the edges are indicated by $A$, and the features of a node are the ROI-level feature vector of the corresponding ROI, i.e., the node feature matrix is the FCN matrix. In the following, we employ a two-layer Graph Convolutional Network (GCN) to smooth the brain graph data, followed by average-pooling to obtain the whole brain-level feature vector on a global scale. The propagation rule of the GCN is defined as

$$\mathbf{H}^{(l+1)} = \hat{\mathbf{A}} \mathbf{H}^{(l)} \mathbf{W}^{(l)} + \mathbf{b}^{(l)}. \tag{2}$$

where $\mathbf{H}^{(l)}$ denotes the feature matrix of the $l$-th layer (with $\mathbf{H}^{(0)}$ being the FCN matrix), $\hat{\mathbf{A}} = \mathbf{D}^{-1/2}(\mathbf{A} + \mathbf{I})\mathbf{D}^{-1/2}$ (where $\mathbf{I}$ is the identity matrix, and $\mathbf{D}$ is the degree matrix associated with $\mathbf{A}$), $\mathbf{W}^{(l)}$ represents the weight matrix of the $l$-th layer, and $\mathbf{b}^{(l)}$ is the bias term.

In the second layer of the GCN, we obtain the refined node feature matrix $\mathbf{H}^{(2)} \in \mathbb{R}^{D \times D}$ (Here the output dimension of the node features is consistent with its original input dimension). The whole brain-level feature vector is then computed by averaging all node features in $\mathbf{H}^{(2)}$, i.e.,

$$\mathbf{f}_{\text{global}} = \frac{1}{D} \sum_{i=1}^{D} \mathbf{h}_i^{(2)}. \tag{3}$$

where $\mathbf{h}_i^{(2)}$ is the $i$-th row of $\mathbf{H}^{(2)}$.

As stated above, we obtain hierarchical features of FCNs which fully bridge FCN information to the LLM. We further concatenate these feature into a unified feature vector, i.e., $\mathbf{f}_{\text{concat}} = [\mathbf{f}_{\text{roi},1}; \mathbf{f}_{\text{roi},2}; \cdots, \mathbf{f}_{\text{roi},D}; \mathbf{f}_{\text{sub},1}; \mathbf{f}_{\text{sub},2}; \cdots, \mathbf{f}_{\text{sub},N}; \mathbf{f}_{\text{global}}] \in \mathbb{R}^{D \times (D+N+1)}$. We apply an MLP to mapping $\mathbf{f}_{\text{concat}}$ to the input space of the LLM, laying the foundation for subsequent alignment between the FCN and text modalities. The mapping process is formulated as

$$\mathbf{f}_{\text{aligned}} = \text{MLP}(\mathbf{f}_{\text{concat}}). \tag{4}$$

## 3.2 Multi-paradigm Task Design for Instruction Tuning

Since multi-task prompt tuning facilitates knowledge sharing across tasks, thereby improving learning efficiency, robustness, and generalization performance (Sanh et al., 2021; Asai et al., 2022; Shen et al., 2024), it provides a strong motivation for designing diverse instruction-tuning data. To this end, at the graph level, we construct tasks from three paradigms—predictive, judgment, and comparative—as illustrated in Figure 1. Through data synthesis, these multi-paradigm tasks enable the LLM to efficiently acquire the knowledge embedded in FCNs. In what follows, we detail these three paradigms, respectively.

**Predictive Paradigm Task.**  The predictive paradigm requires the model to directly predict the corresponding attribute value based on the input FCN and its associated prompt. In this paper, we include 19 attributes (detailed in Appendix A.2), which can be categorized into four types of indicators: (1) Demographic Indicators (Gender, Age, Handedness), (2) Clinical Indicators (mental disease status/diagnosis), (3) Cognitive Function Indicators (FIQ, VIQ, PIQ), and (4) Phenotypic Indicators (personality traits, behavioral measures, and emotional states).

**Judgment Paradigm Task.**  Unlike predictive paradigm task that generates the answer directly, the judgment paradigm task requires the model to assess the consistency between the input FCN and a

given candidate answer, and to output a binary decision accordingly. Specifically, for attributes with a limited set of discrete categories (i.e., categorical attributes), the model is tasked with determining whether a given candidate label matches the input FCN. For continuous attributes, the model assesses whether the true value falls within a predefined interval (e.g., age groups are defined as early childhood, adolescence, early adulthood, middle adulthood, and late adulthood; IQ ranges in 10-point segments). This paradigm can be seen as a coarser-grained binary classification of predictive tasks, with a balanced distribution of positive and negative samples.

**Comparative Paradigm Task.** While the above two tasks enable the model to learn subject-specific representations, they do not capture inter-subject associations within the same task. Contrastive learning has been widely recognized as an effective fashion for acquiring robust and discriminative representations of FCNs (Wang et al., 2022a; 2024d), and its extension to LLMs through self-play has further demonstrated its potential in enhancing alignment and reasoning capabilities (Chen et al., 2024b; Wang et al., 2024b). However, existing methods typically rely on model-specific architectural designs and are not structure-free. To address this limitation, we introduce a comparative paradigm integrating contrastive learning into instruction tuning. For categorical attributes, the model receives two subjects' FCNs and judges if their attribute representations match. For continuous attributes, it compares two subjects' FCNs to identify which has a higher value within a predefined range. Positive and negative pairs are generated from predictive task-associated subject groups with balanced ratios. This paradigm captures inter-subject population relationships, embedding contrastive learning data-drivenly while complementing the single-subject predictive and judgment paradigms above.

### 3.3 MULTI-STAGE LEARNING OF FCN KNOWLEDGE

**Stage I: Pretraining for Robust FCN Knowledge Alignment.** This stage leverages the multi-paradigm instruction tuning data designed in Sec. 3.2 to align the multi-scale embedding tokens of FCNs with the text space of the LLM. During training, we keep the pre-trained LLM frozen and only update the multi-scale FCN encoder, ensuring that the alignment focuses on bridging FCN and text feature spaces without interfering with the LLM's inherent language capabilities.

Notably, since the LLM has no prior exposure to FCN feature data, a sufficient quantity of FCN-involved tasks is required to facilitate effective alignment. For the BOLD time series of a single subject, we can generate multiple time-window-specific FCNs ($\text{FCN}_t$) via a sliding window approach for data augmentation. Specifically, for each ROI, its BOLD time series of length $T$ can be divided into $N_{\text{aug}}$ segments using a sliding window of length $L$ and step size $P$, where

$$N_{\text{aug}} = \left\lfloor \frac{T - L}{P} \right\rfloor + 1. \tag{5}$$

where $\lfloor \cdot \rfloor$ denotes the floor function.

**Stage II: Fine-tuning for High-level FCN Knowledge Refinement.** After the initial pretraining stage, the FCN-LLM acquires basic capabilities to recognize FCNs and capture fundamental FCN-related knowledge. Nevertheless, two key limitations remain. First, training solely on the multi-scale FCN encoder restricts the model's comprehensive understanding of FCNs. Second, the large volume of instruction tuning data using augmented FCN generated via sliding windows introduces inherent noise. To overcome these limitations, the fine-tuning stage incorporates two critical strategies. (1) Training is conducted on high-quality instruction tuning data using original FCNs rather than augmented FCNs, while retaining the same tasks. (2) Both the LLM and the multi-scale FCN encoder are unfrozen and trained jointly, enabling the LLM to more deeply integrate FCN features and allowing the encoder to refine its representations based on high-quality data. Together, these designs enhance the model's high-level understanding of FCN knowledge.

## 4 EXPERIMENTS

### 4.1 EXPERIMENTAL SETTINGS

**Datasets.** We downloaded several publicly available rs-fMRI datasets including HBN (Alexander et al., 2017), HCP (Van Essen et al., 2013), QTIM (Strike et al., 2023), GSP (Holmes et al., 2015),

Table 1: Basic information and usage of the 10 datasets in this study.

| Usage | Name | Size | Available Attributes |
|---|---|---|---|
| Tuning & Internal Test | HBN (Alexander et al., 2017) | 2254 | Gender, Age, and Handness |
| | HCP (Van Essen et al., 2013) | 1080 | Gender, Age, Handness, and Phenotype |
| | QTIM (Strike et al., 2023) | 1024 | Gender, Age, and Handness |
| | GSP (Holmes et al., 2015) | 1569 | Gender, Age, and Handness |
| | ABIDE (Di Martino et al., 2014) | 855 | Gender, Age, Handness, Cognition, and Diagnosis |
| | ADHD (consortium, 2012) | 872 | Gender, Age, Handness, Cognition, and Diagnosis |
| | MDD (Yan et al., 2019) | 2379 | Gender, Age, Handness, and Diagnosis |
| | SRPBS (Tanaka et al., 2021) | 1397 | Gender, Age, Handness, and Diagnosis |
| Zero-shot Test | ABIDE II (Di Martino et al., 2014) | 1044 | Gender, Age, Handness, Cognition, and Diagnosis |
| | CNP (Bilder et al., 2020) | 261 | Gender, Age, and Diagnosis |

ABIDE (Di Martino et al., 2014), ADHD (consortium, 2012), MDD (Yan et al., 2019), SRPBS (Tanaka et al., 2021), ABIDE II (Di Martino et al., 2014), CNP (Bilder et al., 2020). We listed these datasets with basic information and usage in Table 1. Briefly, to construct the diverse and informative instruction-tuning tasks described in Sec. 3.2 for facilitating LLM learning of FCN knowledge, we merged the first eight datasets and split them into training and test sets with an 80/20 ratio. The remaining two datasets were reserved entirely for zero-shot evaluation, simulating real-world scenarios where the model encounters previously unseen data distributions. A total of 19 attributes are included in these datasets, comprising age, gender, handedness, mental disease diagnosis, overall intellectual ability (FIQ), verbal intellectual ability (VIQ), performance intellectual ability (PIQ), and twelve specific phenotypic indicators (detailed in Appendix A.2), as summarized in Table1.

**Data Synthesis.** For each attribute, we generated 50 instruction prompts under different paradigms. From these prompts, one was randomly selected, and the corresponding attribute value was used as the response, forming a question–answer pair for instruction tuning. In the first stage of alignment learning, we generated over 2.3 million instruction-tuning pairs, while in the second stage of high-level knowledge learning, approximately 426,000 pairs were created. The preprocessing of data and the exact number of instruction-tuning pairs per paradigm was detailed in Appendix A.3 and A.4.

**Implementation Details.** We used the AAL atlas to partition the human brain into $D = 116$ structural ROIs. We followed (Yeo et al., 2011) to define $N = 7$ functional subnetworks and assigned the ROIs of the AAL template to these subnetworks according to (Power et al., 2011). It is worth noting that not all ROIs were mapped to the predefined functional subnetworks. A 2-layer GCN extracts global brain level features: its adjacency matrix $\mathbf{A}$ is derived by taking absolute values of connectivity strengths from the FCN, and filtering weak connections with a threshold of 0.5 (Wang et al., 2024d); the GCN's hidden and output layers have dimensions 256 and 116, respectively, with average pooling applied to the output to obtain a single global feature vector. Finally, a multi-scale encoder converts FCNs into 124 FCN tokens (i.e., $D + N + 1$), which replace the special placeholder $\langle \text{FCN} \rangle$ in prompts to form the final FCN-LLM input. Besides, we employ Qwen2.5-3B and 7B Yang et al. (2024a) as the pre-trained LLM for our FCN-LLM. For more details on hyper-parameter settings, training processes and datasets, please refer to the Appendix A.4.

**Evaluation Metrics.** We evaluated our framework on five tasks: gender classification, disease classification, age prediction, cognitive function prediction, and phenotype prediction. Specifically, gender classification is a binary task that distinguishes between male and female subjects. For disease classification, since our dataset includes nine disorders (Autism Spectrum Disorder (ASD), Attention Deficit Hyperactivity Disorder (ADHD), Major Depression Disorder (MDD), schizophrenia (SZ), Obsessive Compulsive Disorder (OCD), etc.), we defined it as a six-class task by categorizing subjects into health controls (HC), ASD, ADHD, MDD, SZ, and a combined group of other psychiatric disorders, according to sample distribution. Age prediction aims to estimate the absolute age of each subject, while both cognitive function and phenotype prediction involve predicting standardized scores. For classification tasks, we report accuracy (ACC), Matthews correlation coefficient (MCC), and F1-score as evaluation metrics, while for regression tasks, we adopt mean absolute error (MAE) and pearson correlation coefficient (PCC). Since both the cognitive function and phenotype prediction

Table 2: Performance of three types of methods on the internal test set.

| Type | Method | Gender Classification | | | Disease Classification | | | Age Prediction | | Cog Prediction | | Pheno Prediction | |
|---|---|---|---|---|---|---|---|---|---|---|---|---|---|
| | | ACC | MCC | F1 | ACC | MCC | F1 | MAE | PCC | MAE | PCC | MAE | PCC |
| Supervised | GCN | 65.59 | 30.14 | 65.53 | 66.86 | 50.26 | 63.37 | 6.197 | 73.13 | 0.097 | 79.11 | 0.146 | 4.75 |
| | HGCN | 63.50 | 27.97 | 59.62 | 66.97 | 50.73 | 63.89 | 5.892 | 74.91 | 0.095 | 78.73 | 0.144 | 5.56 |
| | BrainNetCNN | 64.28 | 29.16 | 62.61 | 65.83 | 49.73 | 64.87 | 6.185 | 76.10 | 0.12 | 72.72 | 0.167 | -2.16 |
| | BrainGNN | 63.86 | 28.40 | 63.89 | 68.11 | 52.99 | 66.43 | 6.062 | 73.67 | 0.091 | 81.42 | 0.144 | 3.57 |
| | Transformer | 69.60 | 38.06 | 69.44 | 68.23 | 52.30 | 65.33 | 5.041 | 80.42 | 0.091 | 79.50 | 0.144 | 0.98 |
| | BNT | 71.91 | 42.80 | 71.77 | 71.43 | 58.48 | 71.52 | 4.621 | 83.79 | 0.090 | 80.75 | 0.143 | 3.28 |
| Foundation | BrainNPT | 64.71 | 28.34 | 63.62 | 68.10 | 52.37 | 64.53 | 7.570 | 57.47 | 0.165 | 50.23 | 0.154 | 1.48 |
| | PTGB | 62.29 | 24.08 | 61.07 | 67.09 | 53.14 | 65.20 | 8.285 | 45.62 | 0.167 | 51.64 | 0.162 | 4.25 |
| | CINP | 63.01 | 23.78 | 61.82 | 67.60 | 51.90 | 64.74 | 6.941 | 54.24 | 0.164 | 53.30 | 0.148 | 6.68 |
| | BrainMass | 67.28 | 33.22 | 67.03 | 69.26 | 55.37 | 63.98 | 7.084 | 62.76 | 0.153 | 74.27 | 0.145 | 5.93 |
| Ours | FCN-LLM (3B) | 65.48 | 29.82 | 65.39 | 70.97 | 57.41 | 70.78 | 6.293 | 75.81 | 0.089 | 76.77 | 0.173 | 6.95 |
| | FCN-LLM (7B) | 66.41 | 32.21 | 66.38 | 69.86 | 54.44 | 68.86 | 6.178 | 76.38 | 0.090 | 77.61 | 0.138 | 6.49 |

Table 3: Performance of three types of methods on the zero-shot test set.

| Type | Method | ABIDEII | | | | | CNP | | |
|---|---|---|---|---|---|---|---|---|---|
| | | Disease Classification | | | Cog Prediction | | Disease Classification | | |
| | | ACC | MCC | F1 | MAE | PCC | ACC | MCC | F1 |
| Supervised | Transformer | 22.77 | 3.59 | 26.73 | 0.154 | -3.00 | 36.74 | 0.16 | 29.78 |
| | BNT | 31.82 | 6.19 | 40.16 | 0.158 | 10.66 | 44.44 | 2.88 | 30.26 |
| Foundation | BrainNPT | 31.56 | 2.21 | 27.59 | 0.147 | 4.32 | 40.71 | 5.23 | 31.43 |
| | PTGB | 32.14 | 3.28 | 29.53 | 0.145 | 3.79 | 38.91 | 1.24 | 28.87 |
| | CINP | 32.66 | 1.34 | 30.67 | 0.130 | 5.93 | 44.44 | 0.12 | 29.31 |
| | BrainMass | 35.88 | 5.65 | 34.04 | 0.121 | 4.85 | 46.36 | 3.19 | 29.93 |
| Ours | FCN token + SVM | 32.16 | 1.65 | 38.41 | 0.133 | 5.37 | 46.43 | 2.07 | 30.02 |
| | FCN-LLM (3B) | 53.75 | 17.69 | 48.66 | 0.093 | 12.47 | 49.43 | 9.71 | 47.62 |
| | FCN-LLM (7B) | 54.03 | 18.23 | 49.15 | 0.091 | 10.26 | 49.31 | 9.26 | 46.72 |

tasks involved multiple attributes, we reported the average results across the different attributes in the corresponding tables.

## 4.2 QUANTITATIVE RESULTS

**Performance Comparison of FCN-LLM and Baselines on the Internal Test Set.** We compared FCN-LLM with six supervised FCN models and four FCN foundation models; detailed descriptions of the baselines were provided in Appendix A.5. As shown in Table 2, the supervised BNT achieved the best performance across all tasks, demonstrating the strong capability of supervised learning when focusing on specific objectives. Notably, our FCN-LLM attained state-of-the-art results among all non-supervised methods, even surpassing them in the MAE metric for cognitive function prediction and in the PCC metric for phenotype prediction. This indicates that leveraging LLMs to understand FCNs can enhance the learning of the overall distribution of FCN data, thereby better capturing global relational patterns among FCNs. Compared to the Qwen2.5-3B-based FCN-LLM, the Qwen2.5-7B-based variant achieved improvements on gender classification and regression tasks but exhibited a slight decline on disease classification. Two factors may account for this observation: (1) increasing the scale of the LLM does not necessarily enhance its ability to understand FCNs; and (2) larger models may require more extensive instruction-tuning data to effectively acquire FCN-related knowledge.

**Performance Comparison of FCN-LLM and Baselines on the Zero-shot Test Set.** We compared FCN-LLM against two supervised FCN models that achieved the best performance on the internal test set, as well as four FCN foundation models. We can observe from Table 3 that FCN-LLM outperformed both FCN foundation models and supervised models on the zero-shot test sets, demonstrating strong generalization capability. This advantage primarily arises from the flexibility of FCN-LLM, which allows querying in arbitrary textual forms. For instance, in the ABIDE II dataset, which only contained two classes, we adapted the prompts by restricting the candidate labels to HC and ASD. In contrast, other models can only transfer the six-class classifier to this binary setting in a zero-shot manner, leading to degraded performance. Compared with Qwen2.5-3B-based FCN-LLM,

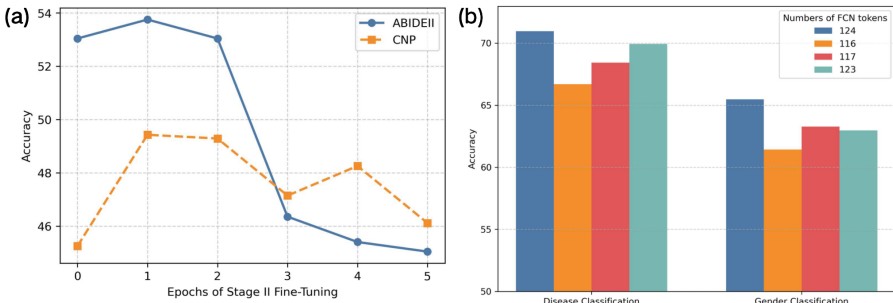

Figure 3: (a) The impact of continual training on the generalization performance of FCN-LLM in Stage2; (b) The ablation study on the proposed multi-scale FCN encoder, which explores the effects of different types of FCN tokens—specifically, combinations of 116 ROI-level tokens, 7 subnetwork-level tokens, and 1 brain-level token.

Qwen2.5-7B-based variant achieved improved results on the ABIDE II dataset but showed a slight performance drop on the CNP dataset, indicating broadly comparable generalization across zero-shot datasets.

### 4.3 ABLATION STUDIES

**Impact of Over-tuning on Generalization.** As shown in Figure 3(a), we illustrated the performance of Stage II joint fine-tuning of the multi-scale FCN encoder and the LLM on the disease classification task of the zero-shot test set. The results indicate that while Stage II fine-tuning initially improves model performance, further increasing the number of fine-tuning epochs leads to a performance decline. It suggests that the model may overfit with the fine-tuning data, thereby reducing its generalization ability on unseen data. To mitigate this, we set the number of fine-tuning epochs in Stage II to one. This finding also highlights the need for future work to enrich the diversity of instruction-tuning data and incorporate larger-scale FCN datasets.

**Effectiveness of Multi-scale FCN Features.** To demonstrate the effectiveness of our multi-scale FCN encoder, we evaluated different feature combinations on the internal test set for both disease and gender classification tasks, as shown in Figure 3(b). The results indicate that incorporating all three scales of features yields the best performance overall. For disease classification, adding functional subnetwork-level features provides larger performance gains compared to adding whole brain-level features, implying that disease discrimination relies more heavily on localized subnetwork information. In contrast, for gender classification, incorporating global brain-level features results in greater improvements than incorporating subnetwork-level features, indicating that gender-related differences are more strongly reflected in global brain patterns. These findings highlight the importance of leveraging multi-scale representations of FCNs to capture task-specific information effectively.

**Ablation for Comparative Paradigm Task.** We conducted additional experiments for the comparative paradigm task proposed in Sec. 3.2, and results detailed in Appendix A.6 demonstrate its effectiveness.

## 5 CONCLUSION

We present FCN-LLM, the first framework to align brain functional connectivity networks with the text modality, enabling LLMs to understand FCNs. By combining a multi-scale FCN encoder with graph-level, multi-task instruction tuning, FCN-LLM learns functional knowledge across brain regions, subnetworks, and whole-brain connectivity. Extensive experiments on multi-site datasets show that FCN-LLM achieves strong zero-shot generalization, outperforming existing supervised FCN models and foundation models on unseen tasks. These results highlight the potential of integrating neuroscience data with LLMs, providing a flexible and interpretable platform for clinical prediction and brain research. Future work may explore incorporating dynamic brain connectivity,

extending to other neuroimaging modalities, and leveraging richer language-based knowledge to further enhance model understanding and generalization.

**Ethics Statement.** The authors have carefully read and adhered to the ICLR Code of Ethics. This work does not involve human subjects, sensitive data, or any research practices that raise ethical concerns. We confirm that all aspects of this research, including data usage, experimental procedures, and reporting, fully comply with ethical standards and conference guidelines. Additionally, there are no conflicts of interest, and all data handling ensured privacy and security in accordance with applicable regulations.

**Reproducibility statement.** We have made extensive efforts to ensure the reproducibility of our work. Details regarding dataset acquisition and preprocessing, including descriptions of the data attributes used, are provided in Appendix A.2 and A.3 . The generation of instruction-tuning data and multi-paradigm tasks is described in Sec. 3.2. Our model architectures, training procedures, and hyperparameter settings for both pretraining and fine-tuning stages are presented in Sec. 3.1, 3.3 and Table 7. Additional implementation details and supplementary analyses can be found in the Appendix A.4. The code and scripts necessary to reproduce the experiments will be made publicly available upon acceptance of the paper.

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

## A APPENDIX

### A.1 FUNCTIONAL CONNECTIVITY NETWORK

FCNs are typically constructed by calculating the correlation of mean BOLD signals between each pair of brain regions (Pearson correlation is adopted in this study) (Friston, 1994). Specifically, assuming the BOLD signal of brain region $a$ is $x$ and that of brain region $b$ is $y$, the Pearson correlation coefficient is computed as follows:

$$r_{x,y} = \frac{\sum_{i=1}^{n}(x_i - \bar{x})(y_i - \bar{y})}{\sqrt{\sum_{i=1}^{n}(x_i - \bar{x})^2}\sqrt{\sum_{i=1}^{n}(y_i - \bar{y})^2}} \qquad (6)$$

where $\bar{x}$ and $\bar{y}$ denote the mean values of $x$ and $y$, respectively, and $n$ represents the number of time points in the BOLD signals.

FCNs characterize inter-regional interaction responses of the brain, representing the human brain at the regional scale. However, existing studies have shown that further dividing brain regions into sub-networks (i.e., grouping brain regions with similar functions) and exploring interactions between these sub-networks is a critical approach for decoding features such as disease-related patterns.

Additionally, numerous studies have demonstrated remarkable potential in downstream tasks by leveraging methods like Graph Neural Networks (GNNs) to learn unified representations of FCNs at the whole-brain scale (Kipf & Welling, 2016).

## A.2 DATASETS

The data distribution in the dataset (original vs enhanced) is shown in Table 4, which consists of five specific parts: Gender Distribution, Handedness Distribution, Age Group Distribution, Diagnosis Distribution, and Phenotype Statistics. In the paper, we conducted the time-window FCN enhancement with the length of sliding window $L$ as 100 and the step size $P$ as 20. Besides, the Phenotype Statistics part includes HCP phenotypes with 12 specific metrics: 1) Visual-Spatial Processing Test, Total Correct, 2) NIH Toolbox Oral Reading Recognition Test, 3) Perceived Stress, 4) Anger - Aggression, 5) NIH Toolbox Grip Strength Test, 6) 2-Minute Walk Test, 7) NIH Toolbox Picture Vocabulary Test, 8) NIH Toolbox List Sorting Working Memory Test, 9) Anger-Hostility, 10) Loneliness, 11) Meaning and Purpose, and 12) NIH Toolbox 9-Hole Pegboard Dexterity Test. For the specific definition and source of these metrics, please refer to Table 5.

(a) Gender Distribution

| Gender | Original | Enhanced |
|---|---|---|
| Male | 4237 | 39002 |
| Female | 3516 | 40548 |

(b) Handedness Distribution

| Handedness | Original | Enhanced |
|---|---|---|
| Right-handed | 4417 | 16987 |
| Left-handed | 296 | 995 |
| Mixed-handed | 240 | 290 |

(c) Age Group Distribution

| Age Interval | Original | Enhanced |
|---|---|---|
| $[0, 10)$ | 1200 | 2293 |
| $[10, 19)$ | 1994 | 7210 |
| $[19, 40)$ | 3686 | 63517 |
| $[40, 65)$ | 761 | 5622 |
| $[65, 100)$ | 112 | 908 |

(d) Phenotype Statistics

| Phenotype Metric | Original | Enhanced |
|---|---|---|
| FIQ | 1206 | 5985 |
| VIQ | 1033 | 6256 |
| PIQ | 1046 | 6249 |
| 12 HCP Phenotype | $\sim 860 \times 12$ | $\sim 48986 \times 12$ |

(e) Diagnosis Distribution

| Diagnosis | Original | Enhanced | Diagnosis | Original | Enhanced |
|---|---|---|---|---|---|
| Health Control | 1432 | 9168 | ASD | 416 | 2727 |
| MDD | 1183 | 8178 | ADHD | 260 | 1477 |
| Schizophrenia | 117 | 901 | Pain | 19 | 171 |
| Bipolar Disorder | 33 | 297 | Dysthymia | 3 | 27 |
| Others | 18 | 162 | - | | |

Table 4: Demographic Characteristics, Diagnosis Distribution and Phenotype Sample Sizes (Original vs. Time-Window Enhanced Dataset)

## A.3 DATA PREPROCESSING

All rs-fMRI data were preprocessed by fMRIPrep (Esteban et al., 2019), an easily accessible, state-of-the-art fMRI data preprocessing pipeline that is robust against variations in scan acquisition protocols. For rs-fMRI data, slice timing correction, head-motion correction, skull-stripping, and spatial normalization to MNI152 space were conducted. After such preprocessing procedures, rs-fMRI data had a voxel size of $2mm^3$. To derive FCNs from rs-fMRI on the automated anatomical labelling (AAL) atlas (Tzourio-Mazoyer et al., 2002) that divide the whole brain into 116 ROIs, we defined FC as Pearson's correlation between BOLD time courses of paired ROIs. To align the node feature dimensions of the graph-represented FCNs from datasets where the scanning durations were different, we employed nodal connection profiles, i.e., the corresponding row for each node in the FCN matrix, as the node features. The defined node features have been demonstrated to

Table 5: Detailed Information of Phenotypic Indicators

| Variable Name | Description | What It Measures | Higher Value Indicates |
|---|---|---|---|
| VSPLOT_TC | Visual-Spatial Processing Test, Total Correct | Visuospatial ability | Better spatial reasoning and more correct responses |
| ReadEng_Unadj | NIH Toolbox Oral Reading Recognition Test | Reading and word recognition ability | Stronger reading and language decoding skills |
| PercStress_Unadj | Perceived Stress | Subjective stress level | Higher perceived stress |
| AngAggr_Unadj | Anger - Aggression (from PROMIS) | Aggressive behavior when angry | Greater tendency toward aggressive responses |
| Strength_Unadj | NIH Toolbox Grip Strength Test | Physical strength (hand grip) | Greater muscular strength |
| Endurance_Unadj | 2-Minute Walk Test | Cardiovascular/physical endurance | Better endurance and longer walking distance |
| PicVocab_Unadj | NIH Toolbox Picture Vocabulary Test | Receptive vocabulary and language | Larger vocabulary and stronger language comprehension |
| ListSort_Unadj | NIH Toolbox List Sorting Working Memory Test | Working memory capacity | Stronger working memory and cognitive processing ability |
| AngHostil_Unadj | Anger - Hostility (from PROMIS) | Hostile attitudes or feelings | Greater sense of hostility |
| Loneliness_Unadj | Loneliness (from PROMIS) | Subjective feelings of loneliness | More intense feelings of social isolation |
| MeanPurp_Unadj | Meaning and Purpose (from NIH Toolbox) | Sense of life meaning and purpose | Stronger sense of purpose and life satisfaction |
| Dexterity_Unadj | NIH Toolbox 9-Hole Pegboard Dexterity Test | Hand and finger motor coordination | Greater manual dexterity and fine motor skills |

Table 6: Details of Cognitive Function Indicators

| Abbreviation | Full Name | Meaning | Description |
|---|---|---|---|
| FIQ | Full-scale IQ | Overall intellectual ability | Reflects a person's overall intelligence level, typically derived from a weighted combination of multiple subtest scores |
| VIQ | Verbal IQ | Verbal intellectual ability | Measures language-related cognitive abilities, such as vocabulary, comprehension, and verbal reasoning |
| PIQ | Performance IQ | Performance (non-verbal) intellectual ability | Measures non-verbal cognitive abilities, such as spatial reasoning, pattern recognition, and hand-eye coordination |

achieve superior performance over other kinds of node features, such as node identities, degrees, and eigenvector-based embeddings (Cui et al., 2022; Kan et al., 2022a).

For continuous attributes, we standardize the output formats and eliminate the influence of varying value ranges by: (1) applying Min–Max normalization to rescale all regression values into [0,1] (He et al., 2024); and (2) linearly mapping the normalized values to integers within [0,100]. Consequently, all regression outputs are represented as integers in [0,100], which are used as standardized FCN-LLM supervision signals, thereby ensuring consistent multi-task output formats.

## A.4 EXPERIMENTAL SETTINGS

We present additional details about our experimental configuration to facilitate the reproduction of our model. The hyperparameters for all stages are summarized in Table 7 and details about the datasets are summarized in Table 8. Additionally, the number of predictive tasks is consistent with that of judgment tasks: each predictive task can be converted into a judgment task by providing a candidate answer and requiring a "yes/no" output. For comparative tasks, we manually define a fixed target quantity (e.g., 50k) for each task, which results in a total of 700K samples—approximately equal to the number of samples for predictive and judgment tasks—in the first training stage; we then reduce this total quantity by half during the second training stage.

During model inference, we flexibly adjusted the prompts based on the attribute type and prior knowledge of the test dataset. For example, in disease classification, candidate disease categories were provided in the prompt and adapted according to the specific disorders present in each test set. Additionally, we employed a self-consistency strategy (Wang et al., 2022b) to aggregate multiple prompt responses, aiming to achieve optimal performance. We employed a keyword-extraction and rule-matching approach to parse the text-based responses generated by FCN-LLM into comparable labels. Outputs that could not be parsed, such as garbled text, were directly counted as incorrect.

| Size | Stage 1 | Stage 2 |
|---|---|---|
| Batch size | 32 | 32 |
| Learning rate | 1e-3 | 1e-5 |
| Epochs | 1 | 1 |
| Learning schedule | Cosine decay | Cosine decay |
| Warm-up ratio | 0.03 | 0.03 |
| Weight decay | 0 | 0 |
| BF16 | ✓ | ✓ |
| TF32 | ✓ | ✓ |
| DeepSpeed stage | ZeRO2 | ZeRO2 |
| GPUs | 8xA100 | 8xA100 |
| Multi-scale encoder | train | train |
| LLM | freeze | train |

Table 7: The hyperparameters for model training.

| Stage | Dataset | | Samples |
|---|---|---|---|
| Stage1 | Enhanced FCN data | Predictive task | 806K |
| | | Judgment task | 806K |
| | | Comparative task | 700K |
| Stage2 | High-level SFT data | Predictive task | 38K |
| | | Judgment task | 38K |
| | | Comparative task | 350K |

Table 8: The dataset details used for model training.

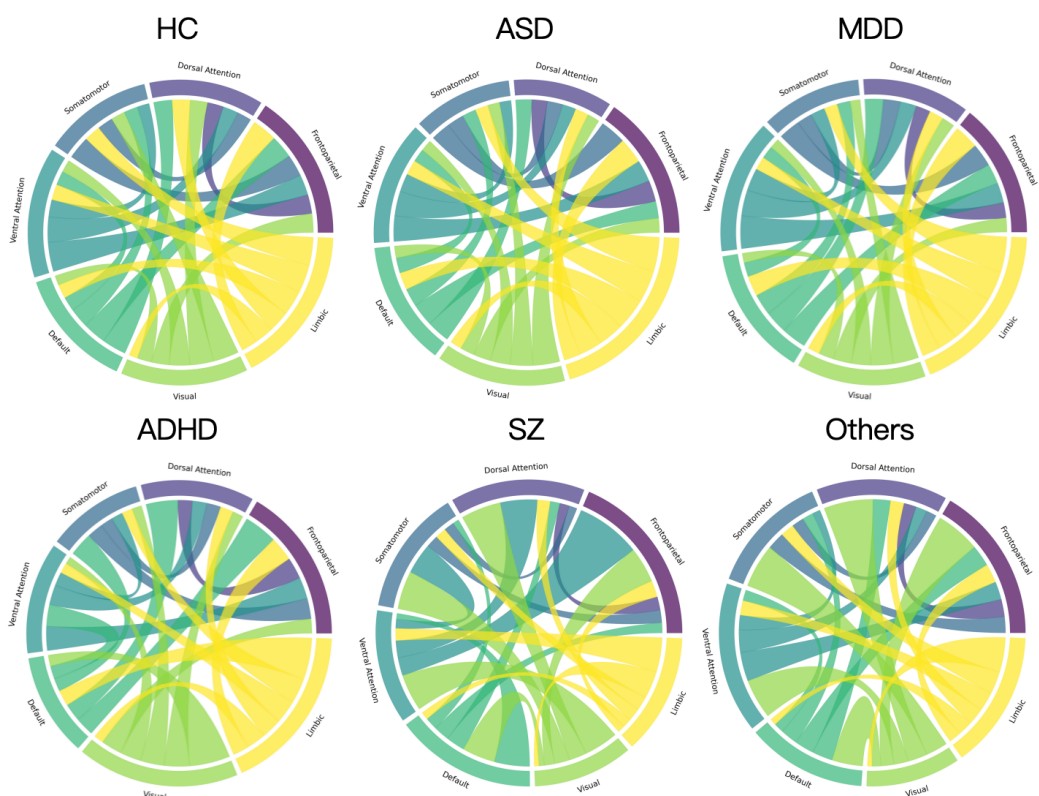

Figure 4: Visualization of connections among FCN tokens of different groups of diseases from the subnetwork perspective.

## A.5 BASELINES

We mainly compared our FCN-LLM with two types of models for brain functional connectivity network analysis: 1) supervised models which are training from scratch containing of GCN (Kipf & Welling, 2016), HGCN (Wang et al., 2024d), BrainNetCNN (Kawahara et al., 2017), BrainGNN (Li et al., 2021), Transformer (Kan et al., 2022b) and BNT (Kan et al., 2022b); 2) foundation models for FCNs including BrainNPT (Hu et al., 2024), PTGB (Yang et al., 2023), CINP (Hu et al., 2025), and BrainMass (Yang et al., 2024b).

For supervised FCN models, we further split the initial training set into training and validation subsets with a 90/10 ratio. Using hyperparameter search, we trained a separate model for each attribute and selected the one with the best performance on the validation set. The final performance was then evaluated on both the internal test set and the zero-shot test sets. For comparison using FCN embeddings, we extracted embeddings of the training and test sets with the base FCN model, normalized the training embeddings, and trained an SVM classifier for each attribute. These classifiers were subsequently applied to the internal and zero-shot test sets to obtain evaluation metrics.

## A.6 ABLATION FOR COMPARATIVE PARADIGM TASK

As shown in Table 9, we specifically explored the impact of including the comparative paradigm task during training on the performance of our proposed FCN-LLM (3B) model. From the results, it can be observed that incorporating the comparative paradigm task—where the model learns relationships between different samples in a data-driven manner during training—facilitates more effective FCN decoding. For instance, this leads to a 2.25% performance improvement on the disease classification task.

Table 9: Ablation study on the comparative paradigm task for FCN-LLM (3B) on the internal test set, evaluated in terms of gender classification, disease classification, and age prediction.

| Comparative task | Gender Classification | | | Disease Classification | | | Age Prediction | |
|---|---|---|---|---|---|---|---|---|
| | ACC | MCC | F1 | ACC | MCC | F1 | MAE | PCC |
| ✗ | 64.23 | 29.03 | 64.89 | 68.72 | 54.69 | 68.85 | 6.631 | 73.28 |
| ✓ | 65.48 | 29.82 | 65.39 | 70.97 | 57.41 | 70.78 | 6.293 | 75.81 |

### A.7 VISUALIZATION FOR DISEASE BIOMARKERS

FCN-LLM fundamentally models maximum likelihood outputs based on FCN feature inputs and text prompts. By conditioning on the reference prompt, the model identifies which parts of the ROI features significantly influence the output. As shown in Figure 4, to detect brain biomarkers associated with disease, we obtained the average attention map across each layer of our FCN-LLM and visualized connections among FCN tokens from the subnetwork perspective.

The process for obtaining the attention map among FCN tokens involves several steps: (1) we select the attention scores between FCN tokens and category outputs (i.e., HC/MDD/ASD, etc.) when prompting for disease diagnosis; (2) we sum the attention scores across both attention head dimensions and category outputs; (3) we map the normalized attention scores onto the FCN tokens to obtain interactions among each FCN token.

We can conclude from the figure that in the ADHD group, resting-state fMRI studies reveal abnormal functional interactions between the default mode network (DMN) and attention networks, potentially impairing attentional regulation (Zhang et al., 2024). In the MDD group, functional connectivity between the visual network (VN) and salience or dorsal attention networks (SN/DAN) is increased, which may reflect heightened sensitivity to external stimuli (Machaj et al., 2024). In the ASD group, the DMN exhibits reduced connectivity, potentially underlying deficits in social cognition and self-referential processing. These findings highlight disorder-specific disruptions in functional network organization, emphasizing the role of network interactions in cognitive and behavioral abnormalities (Hernandez et al., 2015).

### A.8 ATTENTION ANALYSIS

Please see Figure 5 and Figure 6.

### A.9 THE USE OF LARGE LANGUAGE MODELS

In this work, Large Language Models (LLMs) were employed as a general-purpose assistive tool in two primary ways:

**Manuscript Refinement.** LLMs were used to improve the clarity, fluency, and academic style of the manuscript. Specifically, they assisted in rephrasing sentences, correcting grammatical errors, and transforming informal or colloquial expressions into formal, scholarly language. All scientific ideas, experimental design, and interpretations presented in this manuscript were conceived and written by the authors. The LLM did not contribute to the formulation of research hypotheses or the analysis of experimental results.

**Coding Assistance.** LLMs were also used to aid in programming tasks, including writing and refactoring code, debugging, and resolving environment or runtime issues. In particular, they helped with scripting, package installation guidance, and addressing error messages encountered during model implementation. The underlying algorithms, model designs, and experimental procedures were fully developed and implemented by the authors, with the LLM serving purely as a technical assistant.

No part of the LLM-generated content was used as a substitute for original research contributions. Its role was strictly supportive, focusing on improving manuscript presentation and programming efficiency.

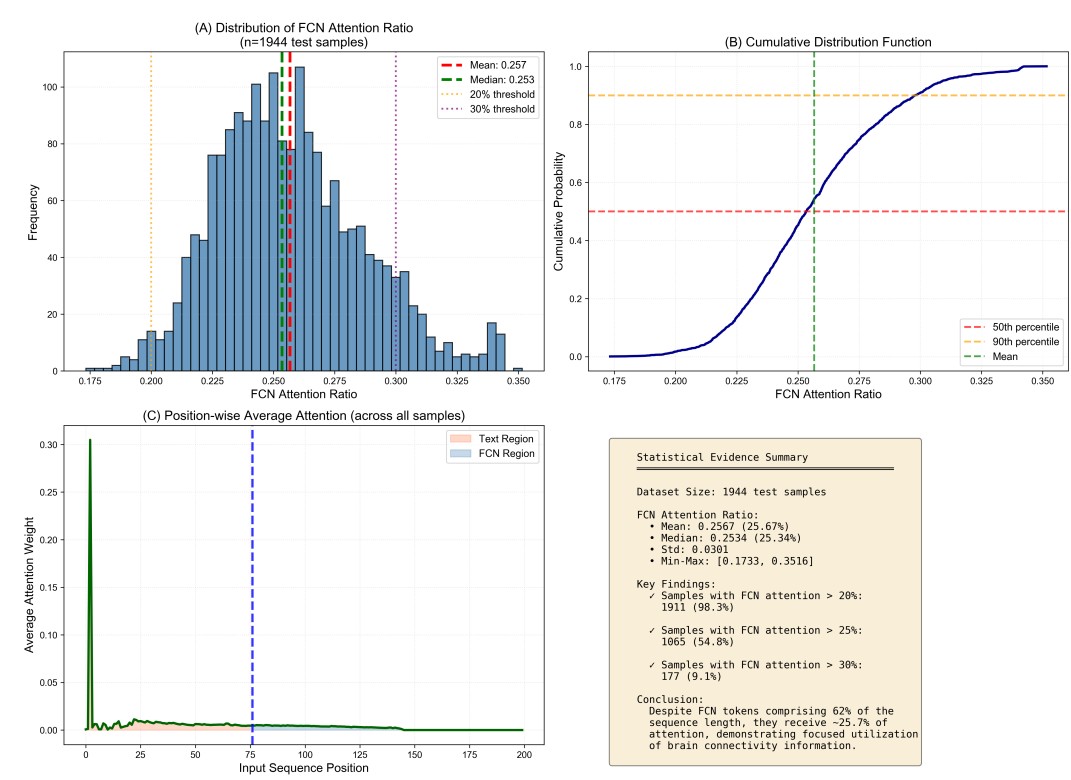

Figure 5: Visualization of statistical evidence comprehensive.

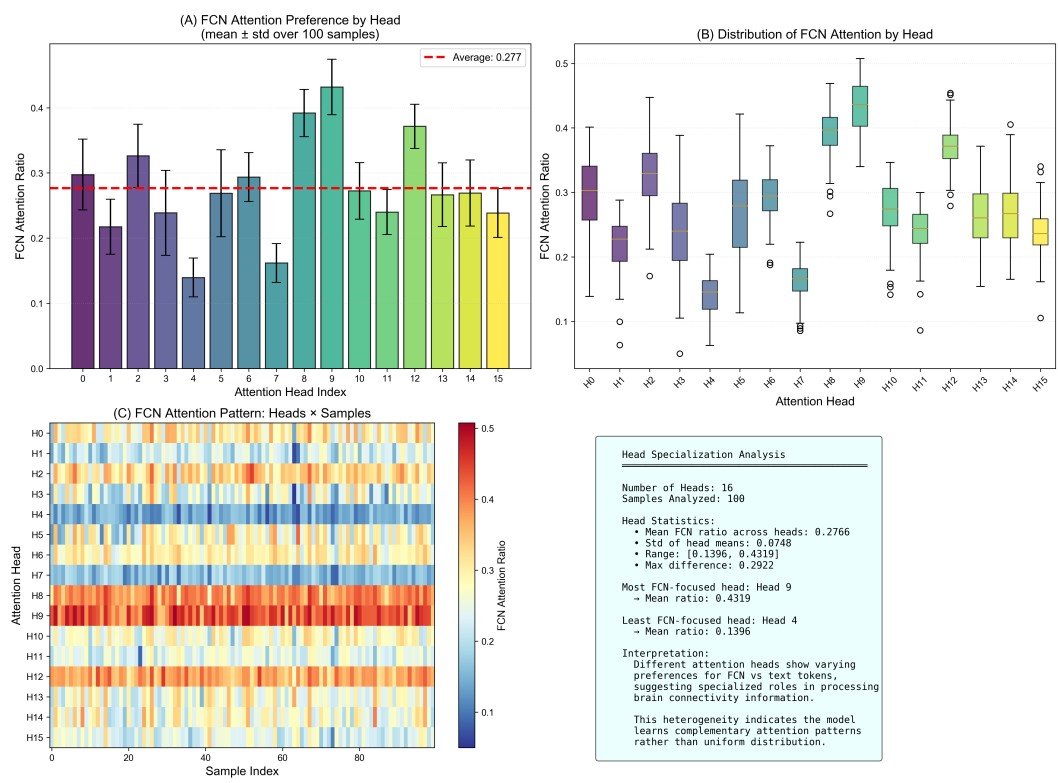

Figure 6: Head specialization analysis.

