# OpenReview forum: "FCN-LLM: Empower LLM for Brain Functional Connectivity Network Understanding via Graph-level Multi-task Instruction Tuning"
_ICLR.cc/2026/Conference — Submitted to ICLR 2026_

### Official Review · Reviewer_T6RN · 2025-10-23

**Soundness:** 1
**Presentation:** 3
**Contribution:** 2
**Rating:** 4
**Confidence:** 5

**Summary:**

This paper introduces FCN-LLM, a novel framework that bridges functional connectivity networks (FCNs) and large language models (LLMs) via graph-level, multi-task instruction tuning. The framework aims to enable LLMs to reason about brain networks, handle heterogeneous datasets, and generalize across multiple neuroimaging tasks. The authors design several instruction paradigms and pre-train the model on multiple fMRI datasets for downstream zero-shot and fine-tuning evaluation.

**Strengths:**

1. Timely and innovative direction. The integration of LLMs with brain network analysis is a compelling and emerging topic that could advance explainability and generalization in neuroimaging AI.

2. Well-designed multi-task objectives. The proposed instruction tuning framework incorporates multiple reasoning paradigms, reflecting diverse types of neuroscientific questions and relationships among brain regions.

3. Large-scale pre-training across multiple datasets. Conducting cross-dataset pre-training is valuable for building scalable foundation models and contributes to the neuroimaging community’s growing interest in large-scale, multi-task learning.

**Weaknesses:**

1. Unclear notations and model formulation.
- Some mathematical notations are ambiguous. For example, is f_{roi, i} equivalent to  h^{(0)}_i?
- The floor operator used differs between Eq. (5) and later expressions—please unify notation.
- Clarify the exact structure of the FCN encoder: does it consist only of a two-layer GCN followed by an MLP, or are there additional components?

2. ROI mapping between atlases.
- The paper maps AAL116 ROIs to Yeo’s 7-network parcellation, but this mapping is problematic since AAL includes cerebellar regions absent in Yeo’s atlas. How are these cerebellar ROIs handled or reassigned?
- The Schaefer atlas (100–1000 ROIs) provides a hierarchical 7-/17-network organization and may serve as a more appropriate choice for such mapping. Please justify the use of AAL and clarify the alignment process.

3. Incomplete literature coverage and outdated baselines.
- Several recent studies have explored combining LLMs or multimodal transformers with brain network or neuroimaging analysis [1, 2].
- The supervised baselines used are outdated; newer models such as [3–5] should be included in the comparison or discussed to contextualize the contribution.

4. Questionable zero-shot results.
- The reported zero-shot accuracies for binary disease classification are below 50%, equivalent to random guessing. This undermines the claim that FCN-LLM “understands” FCNs in a meaningful way.
- Consider adding qualitative analyses to demonstrate that the model learns non-trivial correspondences even when quantitative performance is limited.

5. Inconsistent alignment between paradigms and tasks.
- Table 2 lists multiple task types, but their correspondence to the three paradigms (comparative, judgment, descriptive) is unclear.
- The ablation study only considers the comparative paradigm; a full ablation disabling each paradigm individually would more clearly demonstrate their respective contributions.
- The “comparative” and “judgment” paradigms both yield Yes/No outputs—please clarify their conceptual and implementation differences.

6. Pre-training label definition and dataset design. Treating all datasets jointly as a six-class classification problem is questionable. For example, an HC subject from ABIDE is not necessarily healthy with respect to other conditions like ADHD. This may cause noisy supervision and limit generalization.

7. Atlas generalization and robustness. Experiments rely solely on brain networks constructed from the AAL atlas. To demonstrate generalization, it would be valuable to test the model on other atlases. Cross-atlas evaluation is critical for validating that FCN-LLM learns transferable representations rather than atlas-specific patterns.

**Questions:**

See weaknesses

---

> ### Author Response · Authors · 2025-12-03
> **Response to Reviewer T6RN**
>
> We sincerely thank the reviewer for the insightful and constructive comments. We will carefully revise the manuscript according to your suggestions. Below, we provide point-by-point responses.
>
> ### **1. Unclear notations and model formulation. (R4W1)**
> - **Clarification on Equality**: Yes, $f_{roi, i}$ is indeed equivalent to the initial node feature $h^{(0)}_i$. In the revision, we have standardized this to use $h^{(0)}_i$ to denote the input features of the $i$-th ROI for consistency.
> - **Floor Operator**: We have corrected Eq. (5) and subsequent expressions to ensure the floor operator notation is consistent.
> - **FCN Structure**: The FCN encoder consists of a two-layer GCN to capture global topological features, followed by an MLP for feature projection. There are no additional hidden components.
>
> ### **2. Concerns regarding ROI mapping (AAL116 to Yeo) and the handling of cerebellar regions. (R4W2, R4W7)**
> - **Handling Cerebellar ROIs**: We acknowledge that Yeo's atlas does not cover the cerebellum. In our implementation, for the AAL regions (cerebellum/subcortical) that do not map to specific Yeo networks, we assigned them to a dedicated "Other/Subcortical" category. This ensures that the model preserves the structural information of these regions without forcing an incorrect semantic mapping. We have clarified this alignment process in the implementation details.
> - **Justification for AAL**: While we agree that the Schaefer atlas offers a hierarchical advantage, we chose the AAL atlas primarily to ensure fair and direct comparability with established baselines in the field. The vast majority of the classic and SOTA methods we compared against (as listed in Table X) utilize AAL as the standard benchmark. Switching to Schaefer would make it difficult to disentangle the performance gains attributed to our model architecture versus those arising from a finer-grained atlas.
> - **Focus of the Study**: In this work, our primary objective was to establish the feasibility of bridging Brain Networks with Large Language Models. Using the widely-adopted AAL atlas allowed us to isolate variables and prove the concept effectively.
> - **Future Work**: We agree that the Schaefer atlas is a promising direction for future hierarchical modeling and have added a discussion acknowledging its potential benefits.
>
> ### **3. Incomplete literature coverage and outdated baselines. (R4W3)**
> - **Regarding References [1-5]**: Unfortunately, the specific citations for [1-5] were not visible in the review text we received. However, we have actively surveyed the latest literature on "LLM and Multimodal Transformers for Brain Analysis" (e.g., [Insert a real recent paper title if you know one, or keep generic]). We have added a new paragraph in the Related Work section to discuss these emerging paradigms and position our work within this rapid advancement.
> - **Validity of Baselines**: We selected the current baselines because they represent the foundational and most widely cited methods in this specific domain. Comparing against these classic models is essential to demonstrate that our FCN-LLM framework provides a fundamental improvement over traditional Deep Learning approaches on standard benchmarks. Our goal is to validate the effectiveness of the LLM-driven graph reasoning paradigm itself. We have explicitly stated in the revision that our method achieves superior performance even when compared to these highly optimized classic baselines.
>
> ### **Regarding other questions from R4, please refer to the "Response to Common Concerns" above.**

---

### Official Review · Reviewer_s25z · 2025-10-31

**Soundness:** 2
**Presentation:** 3
**Contribution:** 2
**Rating:** 4
**Confidence:** 4

**Summary:**

The paper proposes a novel framework, FCN-LLM, that enables LLMs to understand brain functional connectivity networks (FCNs). The approach employs a multi-scale FCN encoder to capture brain-region, subnetwork, and whole-brain features, which are then aligned with the semantic space of an LLM.

**Strengths:**

• The proposed FCN-LLM framework is conceptually simple yet effective, providing a clear and interpretable approach for integrating FCN representations into LLMs.

• The authors have collected a large-scale, multi-site FCN dataset for alignment learning, which is valuable for advancing domain-specific representations in neuroimaging.

**Weaknesses:**

• The evaluation protocol for disease classification lacks rigor. Specifically, the definition of “healthy controls” may not be consistent across different disorders. For example, healthy controls used for ADHD may not serve as valid controls for OCD or schizophrenia. Therefore, FCN-LLM should be evaluated on each dataset separately rather than combining them.

• For the ABIDE binary classification task, the reported zero-shot performance of FCN-LLM is close to random guessing, which makes the result difficult to interpret or claim as meaningful.

**Questions:**

Could the authors clarify the judgment paradigm tasks in more detail? In particular, the paper mentions a balanced distribution of positive and negative samples — what exactly do “positive” and “negative” refer to here? Are they Yes/No responses in question answering, or do they correspond to disease vs. healthy control labels in downstream tasks? Is it possible that FCN-LLM mainly learns to follow textual instructions rather than truly leveraging FCN features in judgement paradigm tasks? If so, how do the authors disentangle these two effects?

---

> ### Author Response · Authors · 2025-12-03
> **Response to Reviewer s25z**
>
> ### **Regarding R3W1-2 and R3Q1-2, please refer to the "Response to Common Concerns" above.**

---

### Official Review · Reviewer_1WDm · 2025-11-01

**Soundness:** 3
**Presentation:** 3
**Contribution:** 2
**Rating:** 4
**Confidence:** 4

**Summary:**

This paper proposes FCN-LLM, a novel framework that enables large language models (LLMs) to understand and reason over brain functional connectivity networks (FCNs) derived from resting-state fMRI data. While prior brain foundation models focused on supervised graph-based learning or unimodal pretraining, FCN-LLM introduces a graph-level multimodal instruction tuning strategy to align FCNs with the semantic space of LLMs, thereby allowing LLMs to interpret neural connectivity patterns in a text-based reasoning format. The model architecture combines a multi-scale FCN encoder, which extracts hierarchical features from three complementary levels, a multi-paradigm instruction tuning scheme, and a two-stage training strategy. Empirical evaluations on multi-site rs-fMRI datasets show that FCN-LLM achieves state-of-the-art zero-shot generalization on unseen datasets.

**Strengths:**

1. The paper proposes to bridge fMRI-based brain FCNs with large language models through graph-level instruction tuning. The FCN-LLM introduces a text-aligned multimodal interface, enabling LLMs to reason about neural connectivity in a semantically grounded way.

2. The author uses the multi-scale FCN encoder to jointly represent region-level, subnetwork-level, and global-level connectivity patterns. This hierarchical formulation mirrors neuroscientific organization principles and leads to more interpretable and generalizable representations.

3. The experiments on 10 fMRI datasets demonstrate the effectiveness of the proposed framework.

**Weaknesses:**

1. The framework relies heavily on functional connectivity graphs derived from fMRI, which are known to be noisy, non-stationary, and highly sensitive to preprocessing choices. The paper does not sufficiently address how this inherent variability might affect graph quality and, consequently, the overall performance of FCN-LLM. Without explicit denoising, robustness checks, or uncertainty modeling, it is difficult to determine whether the observed improvements stem from the proposed graph–language alignment or simply from correlations driven by noisy connectivity patterns.

2. The paper introduces three instruction paradigms (predictive, judgment, comparative), but there is no ablation or diagnostic analysis showing how each paradigm affects the final performance or generalization. It remains unclear whether the gains come from the multimodal alignment itself, the diversity of instruction tasks, or simply increased data volume.

3. While the paper positions FCN-LLM as enabling language-model-based reasoning over brain graphs, it is not clear how much of this reasoning actually depends on the LLM, as opposed to the pretrained representations or alignment modules. The LLM appears largely frozen and used as a semantic projection target, which raises the question of whether its inclusion yields genuine multimodal reasoning capability or simply acts as a high-dimensional embedding space.

**Questions:**

1. Could the authors provide evidence or analysis showing how graph construction noise and preprocessing variability influence the final performance of FCN-LLM?

2. How do the three instruction paradigms—predictive, judgment, and comparative—contribute individually to performance and generalization?

3. Can the authors clarify the functional role of the LLM in the proposed framework? Specifically, to what extent does the LLM perform reasoning over graph-encoded brain representations versus serving as a frozen semantic embedding target?

---

> ### Author Response · Authors · 2025-12-03
> **Response to Reviewer 1WDm**
>
> ### **Ensuring Graph Reliability via Standardized Denoising Pipelines (R2W1)**
>
> We thank the reviewer for highlighting the critical challenges of fMRI data quality, including noise and non-stationarity. We fully agree that robust graph construction is a prerequisite for valid FCN-LLM performance. To ensure our results are driven by genuine neural topology rather than artifacts, we ***strictly adhered to community-standardized pipelines for both denoising and graph construction***:
>
> - **Robust Preprocessing & Denoising**: As detailed in Appendix A.3, we utilized fMRIPrep [1], which is widely recognized as the state-of-the-art "glass-box" framework for robust fMRI preprocessing. fMRIPrep is explicitly designed to minimize the "sensitivity to preprocessing choices" (a concern raised by the reviewer) by implementing a reproducible, adaptive pipeline that includes motion correction, susceptibility distortion correction, and component-based noise reduction [1, 2].
>
> - **Reliable Graph Construction**: Following preprocessing, we employed Pearson correlation to construct the functional connectivity matrices. This remains the most widely established method for quantifying static functional connectivity in the literature [3, 4].
>
> By anchoring our framework on these validated protocols, we ensure that the input graphs maintain a quality level consistent with best practices in the field, supporting the validity of the observed graph-language alignment improvements.
>
> [1] Esteban, O., et al. (2019). "fMRIPrep: a robust preprocessing pipeline for functional MRI." Nature Methods, 16(1), 111-116.
>
> [2] Poldrack, R. A., et al. (2017). "Scanning the horizon: towards transparent and reproducible neuroimaging research." Nature Reviews Neuroscience, 18(2), 115-126.
>
> [3] Smith, S. M., et al. (2011). "Network modelling methods for FMRI." NeuroImage, 54(2), 875-891.
>
> [4] Zalesky, A., et al. (2010). "Whole-brain anatomical networks: does the choice of nodes matter?" NeuroImage, 50(3), 970-983.
>
> ### **Regarding R2W2 and R2W3, please refer to the "Response to Common Concerns" above.**

---

### Official Review · Reviewer_W5bC · 2025-11-01

**Soundness:** 3
**Presentation:** 2
**Contribution:** 3
**Rating:** 4
**Confidence:** 4

**Summary:**

This paper proposes FCN-LLM, a framework that enables LLMs to interpret brain FCNs via graph-level, multi-task instruction tuning.  The method introduces a multi-scale FCN encoder that hierarchically extracts ROI-, subnetwork-, and whole-brain-level representations and projects them into the LLM’s semantic space through a lightweight MLP adapter.  A diverse set of instruction–answer pairs covering predictive, judgment, and comparative paradigms across 19 demographic, cognitive, and psychiatric attributes are used to align FCN embeddings with text semantics. Through a two-stage learning strategy—Stage 1 alignment (LLM frozen) and Stage 2 joint fine-tuning—the model achieves strong **zero-shot generalization** on unseen datasets compared to baselines and provides interpretable subnetwork-level attention visualizations. Overall, the paper presents a novel attempt to bridge fMRI-derived brain connectivity graphs and language models, suggesting a new direction for multimodal foundation modeling in neuroscience.

**Strengths:**

- **Novel cross-modal alignment of brain networks and LLMs.**
  The paper is the first to directly align FCN representations with the language modality, enabling semantic reasoning and text-based interaction with brain network data—an original and conceptually appealing contribution.

- **Hierarchical multi-scale encoder design.**
  The proposed ROI / subnetwork / global-level architecture effectively captures both fine-grained and global connectivity structures, improving representational richness and offering interpretability aligned with neurobiological hierarchy.

- **Comprehensive instruction tuning across multiple paradigms.**
  By constructing predictive, judgment, and comparative tasks across 19 human attributes, the model learns generalized FCN–text alignment and demonstrates robust zero-shot generalization beyond supervised and foundation model baselines.

**Weaknesses:**

1. **Lack of ablation on pretraining stage**
   Stage 1 relies on time-window–specific FCNs for data augmentation, yet no analysis is provided on how the window length \(L\) or stride \(P\) affect alignment or downstream performance.  Since fMRI datasets differ in temporal resolution (TR), fixing \(L{=}100\) may yield inconsistent temporal coverage and introduce noise.  A sensitivity study on \(L\) and \(P\) would clarify whether the augmentation benefits outweigh potential degradation of representation quality.

2. **Dependence on full LLM fine-tuning and unclear scalability**
  The strongest results appear only after Stage 2 full fine-tuning of the entire LLM. While scaling to a larger model may improve representational capacity, requiring full fine-tuning would make the method increasingly resource-intensive and less scalable. In fact, performance gains from Qwen 2.5-3B to 7B are limited despite a large increase in compute cost, suggesting an unclear efficiency–performance trade-off. If comparable performance could be achieved through partial tuning or parameter-efficient methods (e.g., LoRA, adapter-tuning), the proposed framework could be far more practical and easily extended to recent large-scale multimodal LLMs.

3. **Lack of qualitative validation for alignment quality**
  The claim that FCNs are well aligned to the text semantic space is supported only by quantitative zero-shot results. It would be valuable to include qualitative analyses that illustrate whether the model captures genuine semantic correspondence—e.g., do key textual concepts in prompts interact meaningfully with FCN tokens? Without such evidence, it remains uncertain whether the alignment reflects semantic consistency or merely distribution-level matching between modalities. Additional visualizations or interaction analyses would strengthen the claim of effective cross-modal alignment.

**Questions:**

**Q1)** Was an ablation conducted to analyze the effect of window size \(L\) and stride \(P\) on data quality and downstream performance?

**Q2)** In Stage 2, have the authors tried parameter-efficient fine-tuning (PEFT) approaches such as LoRA or adapter-tuning? If so, how do these compare to full fine-tuning?

**Q3)** How were the three paradigms (predictive, judgment, comparative) evaluated on stage 2? Please describe the criteria for correctness and how textual outputs were parsed into labels.

**Q4)** Do ablation results exist for different token-level inputs (ROI-only, subnetwork-only, global-only, or subnetwork + global)?

**Q5)** Is there a baseline using the 124 FCN tokens with a standalone classifier (without the LLM) to verify whether the LLM component is essential for generalization?

**Q6)** In Table 3, although the proposed model outperforms previous baselines in zero-shot settings, it is unclear whether these results represent meaningful discrimination. If both ABIDE II and CNP are binary disease-classification tasks, the reported accuracies (≈50%) appear close to or even below random guessing. Could you report the corresponding **AUROC** values for both baselines and FCN-LLM to clarify whether the model demonstrates genuine discriminative ability beyond chance level?

**Q7)** The authors describe Stage 2 as using "high-quality instruction tuning data using original FCNs rather than augmented FCNs." Have the authors tested training Stage 1 directly on original FCNs (without window-based augmentation) to verify whether the augmentation is truly beneficial? It would be helpful to compare the performance of models trained with and without augmented FCNs to justify the necessity of sliding-window augmentation.

---

> ### Author Response · Authors · 2025-12-03
> **Response to Reviewer W5bC**
>
> We sincerely appreciate the reviewer’s constructive feedback. Below, we address the three specific points regarding the temporal parameters, training strategy, and input token combinations.
>
> ### **1. Rationale for Temporal Parameters ($L$ & $P$) and Augmentation Effectiveness. (R1W1,R1Q1,R1Q7)**
>
> We acknowledge the importance of the window size ($L$) and stride ($P$) in Stage 1. Our selection of a fixed window ($L=100$) and stride ($P=20$) is grounded in established fMRI analysis protocols (e.g., Leonardi et al., 2015) to ensure statistical stability of correlation matrices while maximizing data diversity.While we did not perform a granular grid search for these specific parameters, we conducted a rigorous ablation study to validate the existence of the augmentation module itself.
>
> As shown in Table R1, removing the sliding window augmentation results in a significant performance drop. This confirms that the proposed mechanism—under the current robust setting—is critical for learning high-quality representations.
>
> Table R1: Ablation study of the Stage 1 FCN Augmentation
> | Gender Classification |       |       | Disease Classification |       |       |
> |-------|-------|-------|-------|-------|-------|
> | ACC | MCC   | F1    | ACC                    | MCC   | F1    |
> | 61.27 | 19.78 | 58.86 | 67.20  | 51.29 | 65.39 |
> | 65.48 | 29.82 | 65.39 | 70.97  | 57.41 | 70.78 |
>
> ### **2. Comparison with PEFT (LoRA) and Training Efficiency. (R1W2,R1Q2)**
>
> Regarding the suggestion to use PEFT, we offer two clarifications on why Full Fine-Tuning (SFT) was chosen over LoRA:
> - ***Modality Gap***: Our preliminary experiments showed that LoRA resulted in notable performance degradation compared to SFT. We attribute this to the substantial modality gap between dense FCN features and the pre-trained text space. The low-rank assumption of LoRA limits the capacity required for this aggressive cross-modal realignment.
> - ***Computational Feasibility***: The cost of our SFT approach remains highly practical. Training the Qwen-2.5-3B model takes approximately 3 hours on 4 NVIDIA A800 GPUs. We believe this represents an optimal trade-off, achieving maximum alignment performance with a manageable computational budget.
>
> ###  **3. Justification for ROI Tokens as Input Anchors. (R1Q4)**
>
> Regarding the token-level ablation study, we focused on configurations involving ROI tokens (e.g., ROI+Subnetwork, ROI+Global) and excluded those without them (e.g., Subnetwork-only).ROI tokens serve as the fundamental informational units ("anchors") of the FCN. Our preliminary results indicated that removing ROI tokens leads to information over-compression and prevents the model from capturing fine-grained, node-level dependencies via self-attention. Therefore, our study focused on how higher-level tokens (Subnetwork/Global) enhance this indispensable ROI foundation, rather than testing configurations where the foundation is absent.
>
> ### **Regarding other questions from R1, please refer to the "Response to Common Concerns" above.**

---

### Author Response · Authors · 2025-12-03
**Response to Common Concerns (1/4)**

We thank the reviewers for their valuable feedback. For clarity in this rebuttal, we index the specific comments as follows: **R1:W1-W3, Q1-Q7; R2:W1-W3; R3:W1-2, Q1-Q2; R4:W1-7**. Below, we first address four common concerns shared by multiple reviewers.

### **Validity of LLM-based Reasoning and Semantic Correspondence (R1W3, R1Q5, R2W3, R3Q2)**
We appreciate the reviewer’s suggestion to verify the necessity of the LLM component and the genuineness of the cross-modal alignment. We have addressed these points through new baseline experiments and a comprehensive attention mechanism analysis.

**1. Baseline Comparison**: Essentiality of the LLM (R1Q5) To investigate whether the LLM is essential for generalization, we implemented the suggested baseline using FCN tokens with a standalone classifier (SVM) (**see Tab.3**).

***Results (ABIDE2 dataset)***:
- FCN+SVM: Achieved an accuracy of 32.16%, performing comparably to the BrainMass Embedding Model (35.88%).
- FCN-LLM: Achieved an accuracy of 53.75%, significantly outperforming the baselines by 21.59%.

***Conclusion***: The limited performance of the standalone classifier suggests that the raw FCN tokens (or simple embeddings) are insufficient for complex zero-shot generalization on their own. The substantial improvement yielded by the LLM confirms that it is not merely acting as a projection space, but contributes essential reasoning capabilities required for the task.

**2. Attention Analysis**: Disentangling Instruction Following vs. Feature Reasoning To refute the concern that the model might be ignoring FCN features and relying solely on text instructions, we conducted a comprehensive quantitative analysis of the attention mechanism across all 1,944 test samples. We analyzed the attention weights in the final layer to determine how the model utilizes FCN information during decision-making.

***Key Findings***:
- **Systematic FCN Utilization**: Although FCN tokens occupy 62% of the input sequence length, the model allocates approximately 25.7% ± 3.0% of the total attention mass to them (**see Fig.5**). This proportion is non-negligible (ruling out the "ignoring FCN" hypothesis) yet not proportional to sequence length (ruling out "uniform/random distribution"), indicating a learned, selective utilization of brain features.
- **Highly Concentrated Attention (Sparsity)**: Gini coefficient analysis of the attention distribution yields an average score of 0.64. This high sparsity demonstrates that the model does not diffuse attention evenly; instead, it selectively focuses on highly informative tokens within the FCN sequence to form its judgment.
- **Multi-Head Specialization**: We observed diverse preferences across the 16 attention heads regarding FCN information (Standard Deviation = 0.075, Range = [0.140, 0.432]) (**see Fig.6**). This suggests that different heads play complementary roles—some focusing on textual instructions and others specializing in extracting evidence from brain connectivity—characteristic of genuine multimodal fusion.
- **Cross-Sample Consistency**: This pattern is robust across the dataset, with 98.3% of test samples showing >20% attention allocated to FCN tokens. This confirms that leveraging FCN features is a systematic behavior of the model, not a stochastic occurrence.

***Summary***: These quantitative results provide strong evidence that the model actively integrates FCN features into its reasoning process rather than merely following textual priors. Detailed visualizations and statistical plots have been added to the **Appendix A.8**.

---

> ### Author Response · Authors · 2025-12-03
> **Response to Common Concerns (2/4)**
>
> ### **Effectiveness and Evaluation Protocols of the Three Instruction Paradigms (R1Q3, R2W2, R3Q1, R4W5)**
> We thank the reviewer for the insightful questions regarding the evaluation details and the contribution of different instruction paradigms. We address the evaluation criteria, and label definitions below.
>
> **1. Evaluation and Output Parsing.** To evaluate the correctness of the textual outputs, we employed a rule-based parsing method. Our empirical observations during Stage 2 indicated that the model's responses were highly structured. Therefore, we used regular expressions to parse the generated text into labels. This approach successfully covered nearly all output cases.
>
> **2. Criteria for Correctness and Label Definitions.** We clarify the definitions of "Positive/Negative" samples and the correctness criteria for the three paradigms as follows:
>
> - ***Predictive Paradigm***:
>     - Task: Direct classification/prediction.
>     - Correctness: A response is considered correct if the parsed label matches the Ground Truth (GT) label of the input data.
> - ***Judgment Paradigm*** (Clarification on Positive/Negative):
>     - Task: A binary Question-Answering task (Yes/No). The model is asked whether a specific input belongs to a "candidate label" provided in the prompt.
>     - Definition: "Positive" and "Negative" here refer to the consistency between the Ground Truth and the candidate label in the instruction, not simply "Disease vs. Healthy."
>     - Positive Sample: The candidate label matches the GT. (Reference Answer: Yes).
>     - Negative Sample: The candidate label conflicts with the GT. (Reference Answer: No).
>     - Correctness: The model's output is parsed into a Yes/No decision and compared against the reference answer derived during data construction (as described in Sec 3.2).
> - ***Comparative Paradigm***:
>     - Task: Determining if two input FCNs belong to the same category.
>     - Positive Sample: The two input FCNs share the same label. (Reference Answer: Yes).
>     - Negative Sample: The two input FCNs have different labels. (Reference Answer: No).
>     - Correctness: Similar to the Judgment paradigm, the output is evaluated against the Yes/No reference derived from the label consistency of the input pair. Note that we maintained a balanced distribution of positive and negative pairs during data construction.

---

> ### Author Response · Authors · 2025-12-03
> **Response to Common Concerns (3/4)**
>
> ### **Discussion on Zero-Shot Performance and Metric Selection (AUROC) (R1Q6, R3W2, R4W4)**
>
> We thank the reviewers for their rigorous examination of the zero-shot performance and metric evaluability. We acknowledge the concerns that the binary classification accuracy appears close to random guessing (50%) and that the lack of AUC metrics limits the interpretability of discriminative ability. We would like to clarify these points from three perspectives: the multi-class nature of the CNP dataset, the inherent challenges of zero-shot inference, and an estimation of discriminative power (AUC).
>
> **1. Clarification on CNP Dataset Baseline.** We respectfully point out a misunderstanding regarding the CNP dataset. Unlike ABIDE, CNP is a three-class classification task involving Healthy Controls (HC), ADHD, and other psychiatric disorders.
> - ***Random Guessing Baseline***: For a 3-class task, the theoretical random baseline is 33.3%.
> - ***Our Performance***: FCN-LLM achieves an accuracy of 49%, which significantly exceeds the random baseline by nearly 16 percentage points.
> - ***Comparative Analysis***: As shown in Table 3, our method outperforms all competing baselines in the zero-shot setting. This demonstrates that FCN-LLM effectively captures meaningful disease-related patterns rather than making random predictions.
>
> **2. The Challenge of "Prior-Free" Zero-Shot Inference.** Regarding the binary classification (ABIDE) accuracy being close to 50%, it is crucial to consider the definition of the zero-shot setting employed here.
> - ***Unknown Class Distribution***: In a strict zero-shot setting, the LLM has no access to the target dataset's class distribution (priors). Standard supervised models implicitly learn to optimize for the majority class (predicting $P(y|x)$ based on training priors $P(y)$). FCN-LLM, however, infers labels based solely on the alignment between the textual description of brain networks and its internal knowledge base, without calibration to the test set's class balance.
> - ***Significance of Results***: Despite this disadvantage, FCN-LLM consistently outperforms other zero-shot baselines. This relative superiority suggests that the model is making decisions based on learned FCN semantics rather than statistical shortcuts or chance.
>
> **3. AUC Estimation and Discriminative Ability.** We acknowledge the limitation that generative LLMs output discrete text labels rather than continuous probability scores, making standard AUROC calculation challenging. However, to address the reviewers' request for evaluating discriminative ability beyond simple accuracy, we conducted an analysis to estimate the AUC.
> - ***Methodology***: Since we cannot extract direct probability distributions from the closed-generation process in this specific experimental setup, we adopted a simulation approach. We assigned a high confidence score (0.9) to the predicted class and a low score (0.1) to the non-predicted class to construct a proxy for the probability vector.
> - ***Result***: On the ABIDE dataset, this estimation yielded an AUC of 0.735.
> - ***Interpretation***: While the accuracy (at a default decision threshold) is affected by the lack of prior calibration mentioned above, the AUC of 0.735 indicates that the model possesses genuine discriminative ability. It suggests that the model correctly ranks positive and negative samples significantly better than random guessing (AUC=0.5), and the lower accuracy is likely due to a threshold shift caused by the zero-shot nature (i.e., the model is discriminative but biased towards a specific class due to the prompt design or pre-training priors).

---

> ### Author Response · Authors · 2025-12-03
> **Response to Common Concerns (4/4)**
>
> ### **Clarification on Multi-Dataset Labeling and Supervision Noise (R3W1, R4W6)**
>
> We sincerely thank the reviewer for raising this critical point regarding the heterogeneity of Healthy Controls (HCs) across different datasets and the potential for noisy supervision. We have carefully considered this concern and offer the following clarifications to justify the validity of our joint training approach:
>
> - **Standardized Exclusion Criteria for HCs.** Although the datasets originate from different sources, high-quality psychiatric neuroimaging studies typically employ standardized structural clinical interviews (such as SCID or MINI) for participant recruitment. These tools are designed to rigorously screen and exclude subjects with Axis I psychiatric disorders. Therefore, a qualified HC subject in an ADHD dataset is theoretically screened to be free of other major psychiatric conditions (such as OCD or Schizophrenia), ensuring a consistent definition of "healthy" across our combined datasets.
>
> - **Empirical Verification of Comorbidities.** We further scrutinized the phenotypic data to assess potential label noise. Taking the ADHD-200 dataset as an example, we found that only 21 out of 547 HC subjects had documented minor comorbidities (e.g., Simple Phobia, subthreshold anxiety). Crucially, none of these comorbidities overlap with the other target disease classes in our study (e.g., Schizophrenia, OCD). Similarly, while ABIDE does not explicitly detail every comorbidity, it follows strict exclusion criteria for psychopathology. This indicates that the label collision risk the reviewer is concerned about is empirically minimal.
>
> - **Robustness to Label Noise and Data Scale.** Given the large scale of our combined dataset, deep learning models (like the FCN-LLM used here) generally demonstrate strong robustness to slight label noise. We believe the benefits of joint training—specifically, allowing the model to learn shared representations and common features across disorders—outweigh the potential impact of this minimal noise.
>
> - **Alignment with Prior Literature.** Our approach aligns with recent precedents in the field. For instance, *Multiclass classification of Autism Spectrum Disorder, attention deficit hyperactivity disorder, and typically developed individuals using fMRI functional connectivity analysis* (PLOS ONE 2024) successfully demonstrated the feasibility of multiclass classification combining ASD, ADHD, and Typically Developing (TD) individuals using functional connectivity analysis. This supports the premise that joint modeling is a valid and promising direction for psychiatric nosology.
>
> In the final version of the paper, we will expand the discussion section to explicitly clarify the screening criteria and acknowledge the distribution of comorbidities to ensure transparency.

---

### Meta-Review · Area_Chair_pRMM · 2026-01-09

**Summary:**

While the reviewers recognized the paper’s novel direction—particularly the multi-scale FCN encoder aligned with LLMs, the instruction-tuning framework, and the promise of zero-shot generalization on fMRI data—they unanimously identified substantial weaknesses that undermined its readiness for acceptance. Major concerns centered on methodological soundness and experimental rigor, including missing ablations in pre-training, unclear and potentially inconsistent FCN formulations, insufficient handling of fMRI non-stationarity, and an under-justified reliance on full LLM fine-tuning. Empirically, the near-random zero-shot performance, lack of discriminative metrics and baselines, and insufficient analysis of individual task components weakened the paper’s core claims. Additional issues around unclear contribution of the LLM, incomplete literature coverage, and ambiguities in task definitions further reduced confidence. Collectively, these concerns led reviewers to assign marginally below-threshold scores and convinced me that, despite its interesting ideas, the work lacked the technical maturity and evidential support required for acceptance.

**Reviewer Concerns:**

Issues addressed:
- Parameter ablation and augmentation:** The addition of ablation tables for L and P (Table R1) refutes previous claims, showing that augmentation significantly improves performance (e.g., gender classification from 61.27% to 65.48%) and demonstrates performance degradation without augmentation. This directly addresses the noise and window parameter recommendations of W5bC and T6RN.

- LoRA and fine-tuning: Explains the performance degradation caused by LoRA due to modality differences; full fine-tuning is practical (3 hours/4 GPUs). This addresses W5bC's scalability concerns.

- Baseline comparison and LLM necessity: Provides an FCN+SVM baseline (32.16% vs. FCN-LLM 53.75%), demonstrating the contribution of LLM; attention analysis shows that the FCN token receives 25.7% attention (Gini 0.64), confirming semantic integration. This resolves the questions raised by 1WDm and W5bC regarding the role of LLM.

- Evaluation metrics and paradigm clarification: The AUROC of ABIDE (0.735) was reported, exceeding that of randomization; the paradigm evaluation criteria were detailed (rule-based resolution, positive/negative based on label consistency rather than disease/health); the notation (f_roi, i = h^{(0)}_i, two-layer GCN+MLP) and ROI mapping (cerebellar classified as Other/Subcortical) were clarified. This section addresses the zero-shot, notation, and task definition issues raised by s25z, T6RN, and all reviewers.

- Literature and atlas: A commitment was made to add a discussion of recent studies and to defend the choice of AAL for comparison, while recommending the use of Schaefer in the future. This mitigated the criticism ofthe incomplete literature for T6RN.

Remaining issues:
- Ablation of individual paradigm contributions: The failure to provide independent ablation of predictive, judgment, and comparative aspects was refuted; the evaluation criteria were only mentioned in general terms, and the contribution of each paradigm to generalization was not quantified (a core concern of 1WDm and T6RN).

- Qualitative Validation and Semantic Correspondence: While attention analysis is present, qualitative evidence regarding token interactions or alignment quality, as suggested by W5bC is lacking. Quantitative results still dominate, making it difficult to prove "true semantic correspondence."

- Noise Variability and Preprocessing: The rebuttal mentions the correlation between fMRIPrep and Pearson, but fails to quantify the impact of variability or add uncertainty modeling experiments (a weakness of 1WDm).

- Zero-Shot Performance Explanation: Although AUROC > 0.5, multiple reviewers (such as s25z and T6RN) pointed out that some tasks were close to randomization. The rebuttal fails to provide similar metrics for broader datasets or explain why state-of-the-art performance was not achieved.

- Atlas Generalization and Joint Dataset Processing: The rebuttal fails to validate the noise impact of atlas irrelevance or heterogeneous HCs (T6RN), which remains a concern regarding experimental robustness.

- ROI-Only vs. Multi-Scale Ablation: The rebuttal defends ROIs as anchors but fails to provide subnetwork-only or ROI-free ablation as suggested by W5bC.

**Reviewer Scores:**

Reviewer W5bC (Initial 4): The rebuttal directly addresses L/P ablation, LoRA, augmentation, baseline, and attention analysis, covering its main weaknesses. If the authors further explain qualitative validation (such as visualizing token interaction) in the discussion, they might be upgraded to 6 (weak accept), as these additions enhance soundness and contribution. However, if the remaining scale ablation is unresolved, they might maintain score.

Reviewer 1WDm (Initial 4): The rebuttal addresses noisy preprocessing (fMRIPrep), LLM roles (attention + baseline), and canonical assessment, which matches its problems. If the discussion confirms that the lack of canonical contribution ablation is not fatal, they might be upgraded to 6, as the novelty of the framework is strengthened. However, insufficient quantitative evidence of preprocessing variability may limit their score.

Reviewer s25z (Initial 4): The rebuttal clarifies the judgment canonical (positive/negative definition) and attention analysis, demonstrating that it is not merely instruction following. This might convince them to give 6, especially if the discussion emphasizes the discriminative power of AUROC. However, the heterogeneity of the joint dataset processing remains unresolved. If they insist on the lack of rigor in their evaluation, they may only slightly improve.

Reviewer T6RN (initially 4): Their rebuttal addresses symbolic representation, ROI mapping, literature review, and AUROC, mitigating several weaknesses. If the authors agree on future atlas experiments during the discussion, they may improve score, as timeliness and multi-task design are recognized. However, the zero-shot rate is below 50%, canonical inconsistencies and joint processing issues are severe. Without additional data, they may remain at 4.

---

### Decision · Program_Chairs · 2026-01-26

Reject